# Role of *UeMsb2* in Filamentous Growth and Pathogenicity of *Ustilago esculenta*

**DOI:** 10.3390/jof10120818

**Published:** 2024-11-25

**Authors:** Wanlong Jiang, Yingli Hu, Juncheng Wu, Jianglong Hu, Jintian Tang, Ran Wang, Zihong Ye, Yafen Zhang

**Affiliations:** 1Key Laboratory of Microbiological Metrology, Measurement & Bio-Product Quality Security, State Administration for Market Regulation, College of Life Sciences, China Jiliang University, Hangzhou 310018, China; wljiang0112@163.com (W.J.); ylhu0925@163.com (Y.H.); wujun461@gmail.com (J.W.); erdangu8@gmail.com (J.H.); jintiantang@cjlu.edu.cn (J.T.); zhye@cjlu.edu.cn (Z.Y.); 2China National Research Institute of Food and Fermentation Industries, Co., Ltd., Building 6, Yard 24, Jiuxianqiao Middle Road, Chaoyang District, Beijing 100015, China; wran87@outlook.com

**Keywords:** *Ustilago esculenta*, *UeMsb2*, filamentous growth, pathogenicity

## Abstract

*Ustilago esculenta* is a dimorphic fungus that specifically infects *Zizania latifolia*, causing stem swelling and the formation of an edible fleshy stem known as jiaobai. The pathogenicity of *U. esculenta* is closely associated with the development of jiaobai and phenotypic differentiation. Msb2 acts as a key upstream sensor in the MAPK (mitogen-activated protein kinase) signaling pathway, playing critical roles in fungal hyphal growth, osmotic regulation, maintenance of cell wall integrity, temperature adaptation, and pathogenicity. In this study, we cloned the *UeMsb2* gene from *U. esculenta* (GenBank No. MW768949). The open reading frame of *UeMsb2* is 3015 bp in length, lacks introns, encodes a 1004-amino-acid protein with a conserved serine-rich domain, and is localized to the vacuole. Expression analysis revealed that *UeMsb2* is inducibly expressed during both hyphal growth and infection processes. Deletion of *UeMsb2* did not affect haploid morphology or growth rate in vitro but significantly impaired the strain’s mating ability, suppressed filamentous growth, slowed host infection progression, and downregulated the expression of *b* signaling pathway genes associated with pathogenicity. Notably, the deletion of *UeMsb2* did not influence the in vitro growth of *U. esculenta* under hyperosmotic, thermal, or oxidative stress conditions. These findings underscore the critical role of *UeMsb2* in regulating the pathogenicity of *U. esculenta*. This study provides insights into the interaction between *U. esculenta* and *Z. latifolia*, particularly the mechanisms that drive host stem swelling.

## 1. Introduction

*Ustilago esculenta* is a dimorphic fungus belonging to the Basidiomycota phylum, genus Ustilago. It is closely related to *Ustilago maydis* and serves as a significant nutritional endophyte for *Zizania latifolia*, which is currently its only known host [1,2,3,4]. Upon successful infection, *U. esculenta* suppresses flowering in *Z. latifolia* and induces stem swelling, resulting in the formation of edible fleshy stems known as jiaobai [3,5]. In addition to the commercially valuable white jiaobai, two other distinct phenotypes are observed in the field: the male jiaobai, characterized by non-swollen stems, and the gray jiaobai, which has swollen stems filled with teliospores and is inedible [6]. Phenotypic differentiation in jiaobai is closely linked to variations in the pathogenicity of *U. esculenta strains* [4,7]. Furthermore, environmental factors such as nutrients, moisture, and temperature significantly influence the hyphal growth of *U. esculenta* in vitro and its infection proliferation in vivo, leading to either abnormal stem swelling [8,9] or the excessive production of teliospores. Thus, understanding the mechanisms regulating the pathogenicity of *U. esculenta* is essential for elucidating the formation and differentiation of jiaobai phenotypes.

Previous studies have shown that the pathogenicity of *U. esculenta* is primarily characterized by its dimorphic transitions and ability to form teliospores [10,11]. Similar to *U. maydis*, the dimorphic transition in *U. esculenta* involves a shift from the budding of haploid strains to the filamentous growth of dikaryotic mycelium. This process is initiated by the recognition of pheromones and their receptors encoded by the *a* mating-type locus in two sexually compatible haploids, activating the mitogen-activated protein kinase (MAPK) and cyclic adenosine monophosphate-protein kinase A (cAMP/PKA) signaling pathways [9,12,13,14]. These pathways lead to the phosphorylation and activation of the HMG (hydroxymethylglutaryl) transcription factor Prf1 (pheromone response factor 1), resulting in G2 cell cycle arrest, synchronization of mating haploid cells, and the formation and fusion of conjugation tubes to generate dikaryotic mycelium [15]. Additionally, the *b* mating-type locus, which encodes the alleles bE (b-East) and bW (b-West), forms a heterodimeric transcription factor. Prf1 directly regulates the expression of *b* genes by binding to the pheromone response element (PRE) within the *b* mating-type locus. This regulation maintains cell cycle arrest, promotes filamentous growth, and enables subsequent infection processes, completing the dimorphic transition [15,16,17,18]. Over 300 *b* signaling pathway genes have been identified in smut fungi, playing critical roles in hyphal growth, cell cycle regulation, and infection. In *U. esculenta*, several key genes regulated by *b* genes have also been characterized, including *Prf1* (pheromone response factor 1), *Clp1* (clampless1), *Rbf1* (regulator of *b*-filament 1), *Hdp1* (homeodomain protein 1), *Biz1* (*b*-dependent zinc finger 1), and *Kpp6* (kinase pheromone pathway 6), all of which contribute to the modulation of fungal pathogenicity [1,14,19,20]. Teliospores, the thick-walled melanized spores formed after extensive proliferation of dikaryotic hyphae within the host, emerge following nuclear fusion, cell cycle activation, and hyphal detachment. This process is primarily governed by the WOPR transcription factor Ros1 (regulator of sporogenesis 1), which counteracts *b* gene-mediated cell cycle arrest to facilitate teliospore development [21,22,23].

Msb2 (multicopy suppression of a budding defect) was first identified in 1989 as a multicopy suppressor of the budding defect in the cdc24ts mutant of *Saccharomyces cerevisiae* [24,25]. Msb2 serves as a critical upstream sensor protein in the MAPK signaling pathway, capable of detecting external signals and activating MAPK cascades, including the HOG (high-osmolarity glycerol) and FG (filamentous growth) pathways [26,27]. This protein plays a pivotal role in various aspects of fungal biology, including filamentous growth, osmotic regulation, maintenance of cell wall integrity, temperature adaptation, and pathogenicity [28,29,30].

Some research has found that Msb2 functions as a sensor in fungi, exhibiting significant diversity and complexity, and in some fungi, it has overlapping functions with the Sho1 sensor, jointly regulating processes such as filamentous growth, stress response, and pathogenicity. In *Candida albicans*, Msb2 participates in regulating hyphal formation and attachment by modulating the kinase Cek1 activity [28,31]. In *Aspergillus flavus*, Msb2 responds to cell wall stress by positively regulating the phosphorylation of MAP kinases, and *Msb2* mutants display cell wall defects and sensitivity to the cell wall inhibitor caspofungin [32]. In *Arthrobotrys oligospora, Hog1* and *Msb2* mutants were highly sensitive to high osmolarity [33]. In *Fusarium oxysporum*, Msb2 induces the phosphorylation of the MAP kinase Fmk1 and participates in regulating infection and pathogenicity-related functions while also promoting fungal invasive growth [34]. Furthermore, in *U. maydis* and *Magnaporthe oryzae*, Msb2 and Sho1 have overlapping functions in recognizing plant surface signals, activating the Kss1-MAPK pathway, forming appressoria, and promoting pathogenicity [35,36]. In *F. oxysporum*, Msb2 and Sho1 can promote hyphal growth and infection, affecting the integrity of the cell wall [37]. In *C. albicans*, Msb2 and Sho1 cooperate through the homologous gene *Cek1* of the Kss1-MAPK pathway to regulate cell wall formation [28].

To investigate the function and underlying mechanisms of Msb2 in *U. esculenta*, we employed a comprehensive approach. First, we cloned the *UeMsb2* gene, allowing for a detailed analysis of its expression patterns. Additionally, we constructed *UeMsb2* deletion mutants to examine its roles in haploid growth, filamentous growth, infection, and stress responses in *U. esculenta*. This study aims to elucidate the regulatory role of *UeMsb2* in the pathogenicity of *U. esculenta*, providing valuable reference data for further exploration of the mechanisms underlying the pathogenicity of *U. esculenta* and its interactions with *Z. latifolia*. These findings contribute to a deeper understanding of the complex relationships between fungi and their hosts, offering potential insights for disease management or the beneficial exploitation of fungi–host interactions.

## 2. Materials and Methods

### 2.1. Strains, Plants, and Growth Conditions

The wild-type strains used in the experiments were the sexually compatible haploid strains UeT14 (a1b1 CCTCC AF 2015016) and UeT55 (a2b2 CCTCC AF 2015015), isolated and preserved from the gray jiaobai of the cultivated variety “Longjiao No. 2.” The *Escherichia coli* strain used for cloning was DH5α (TaKaRa, Kyoto, Japan). *U. esculenta* was cultured on YEPS medium (1% yeast extract, 2% peptone, 2% sucrose, and solid medium supplemented with 1.5% agar) at 28 °C. Protoplasts of *U. esculenta* were cultured on regeneration medium (1% yeast extract, 0.4% peptone, 0.4% sucrose, 18.22% sorbitol, and solid medium supplemented with 1.5% agar) at 28 °C. *E. coli* was cultured on LB medium (0.5% yeast extract, 1% peptone, 1% NaCl, and solid medium supplemented with 1.5% agar) at 37 °C.

The *Z. latifolia* plants tested were wild *Z. latifolia*, grown in a greenhouse under the following conditions: 25 °C with 12 h of light and 20 °C with 12 h of darkness, at 70% humidity.

### 2.2. Gene Cloning and Bioinformatics Analysis

The genomic DNA of *U. esculenta* was extracted using the CTAB (Cetyltrimethylammonium Bromide) method [38]. The total RNA of *U. esculenta* was extracted using a column-based fungal RNA extraction kit (Sangon Biotech, Shanghai, China). cDNA was synthesized using the HiScript^®®^Ⅱ 1st Strand cDNA Synthesis Kit (+gDNA wiper) (R212-01/02, Vazyme, Nanjing, China). Using the amino acid sequence of Msb2 from *U. maydis* as a query, a BlastP search was performed in the *U. esculenta* genomic database (JTLW00000000) to obtain the predicted genomic sequences and CDS sequences. Specific primers for amplifying the *UeMsb2* genomic and CDS sequences were designed using Primer3.0 (primer synthesis was performed by Youkang, Hangzhou, China.) (Appendix A). PCR amplification was conducted using 2 × Vazyme LAmp Master Mix (P312, Vazyme, Nanjing, China) according to the manufacturer’s instructions. Gel recovery and purifycation were performed using HiPure Gel Pure DNA Kits (Majorbio, Shanghai, China). The T-vector ligation was carried out using the pMD^®®^19-T Vector Cloning Kit (TaKaRa, Kyoto, Japan), and positive transformants were selected and sequenced after transformation into *E. coli* (sequencing work was completed by Youkang, Hangzhou, China.).

The sequencing results were imported into the NCBI ORF Finder tool to determine the open reading frame of the target gene. Important parameters such as molecular weight, isoelectric point, and hydrophobicity of UeMsb2 were predicted using the ExPASy website (https://web.expasy.org, accessed on 15 January 2020). The protein’s conserved domains were analyzed using the NCBI Conserved Domain tool to identify regions that are highly conserved during evolution. The presence of transmembrane regions was predicted using the TMHMM 2.0 tool. The position and size of introns within the gene were confirmed using Clone Manager 8 software. A phylogenetic tree was constructed using MEGA 7.0 software employing the neighbor-joining method.

### 2.3. Construction of Deletion Strains

The deletion strains were constructed following the genetic modification strategy and PEG-mediated protoplast transformation method established by Yu (2015) [39] for *U. esculenta*. First, genomic DNA from UeT14 was used as a template to PCR-amplify the upstream and downstream fragments of the *UeMsb2* gene, while the resistance gene fragment was amplified from the plasmid pUMa1507, which contains the hygromycin resistance gene. The amplified ~1000 bp upstream and downstream fragments of the *UeMsb2* gene were connected with the resistance gene fragment through PCR, resulting in two long fragments that included the resistance gene and the target gene’s flanking sequences. These connected fragments were then cloned into the PMD19-T vector using the ClonExpressⅡ One Step Cloning Kit (C112-01, Vazyme, Nanjing, China), creating a recombinant plasmid containing the resistance gene and the homologous arms of the *UeMsb2* gene. The recombinant plasmid was transformed into the protoplasts of the wild-type strains using the PEG3550/CaCl_2_-mediated protoplast transformation method. Transformants were cultured on regeneration media containing hygromycin to select for strains that successfully integrated the resistance gene. Further screening was performed using PCR and qRT-PCR to validate the *UeMsb2* gene deletion strains UeT14△*UeMsb2* and UeT55△*UeMsb2*. All primers used are listed in Appendix A.

### 2.4. Construction of Expression Vectors and Subcellular Localization

Using cDNA from the UeT14 strain as a template, the coding sequence (CDS) of the *UeMsb2* gene was amplified by PCR. The pUMa932 plasmid, which contains the *cbx* resistance gene, was digested with the restriction enzyme BamHⅠ. The amplified *UeMsb2* gene fragment was ligated with the vector fragment using the ClonExpressⅡ One Step Cloning Kit (C112-01, Vazyme, Nanjing, China). The recombinant plasmid was then transformed into *E. coli*. Positive colonies were screened using the primers pUMa932-F and pUMa932-R for verification. Successfully verified clones were sequenced and preserved. Ultimately, the pUMa932-*UeMsb2*-eGFP expression vector was successfully constructed. Using the PEG3550/CaCl_2_ method, this expression vector was transformed into the protoplasts of the UeT14 strain. Transformants were cultured on a regeneration medium containing 5 μg/mL carboxin to select for successful integration of the resistance gene. The expression of the eGFP fluorescent protein was confirmed under a fluorescence microscope, enabling the identification of UeT14::*UeMsb2*-eGFP expression strains. These expression strains were cultured in YEPS liquid medium for 24 h, followed by staining with the membrane dye FM4-64 (20 μg/mL) for 1 min. The subcellular localization of the UeMsb2 protein was then observed using a TCS-SP8 confocal microscope (Leica Microsystems, Wetzlar, Germany). All primers used in this study are listed in Appendix A.

### 2.5. Construction of Complementation Strains

Using genomic DNA from UeT14 as a template, the *UeMsb2* gene sequence with ~2000 bp upstream fragments was amplified by PCR. The pUMa932 plasmid containing the *cbx* resistance gene was digested with the restriction enzymes KpnⅠ and NcoⅠ. The *UeMsb2* gene fragment and the vector fragment were ligated using the ClonExpressⅡ One Step Cloning Kit (C112-01, Vazyme, Nanjing, China), and transformed into *E. coli*. Verified transformants were sent for sequencing and preservation, ultimately resulting in the complementation vector. The complementation vector was transformed into the protoplasts of UeT14△*UeMsb2* and UeT55△*UeMsb2* strains via PEG3550/CaCl_2_-mediated protoplast transformation after being linearized. The transformants were cultured on a regeneration medium containing 5 μg/mL carboxin to select for strains with successfully integrated resistance genes. Further PCR screening and qRT-PCR validation were conducted to obtain the *UeMsb2* complementation strains UeT14△*UeMsb2*::*UeMsb2* and UeT55△*UeMsb2*::*UeMsb2*. All primers are listed in Appendix A.

### 2.6. Observation of Haploid Morphology and Growth Rate Measurement

The strains UeT14, UeT55, UeT14△*UeMsb2*, UeT55△*UeMsb2*, UeT14△*UeMsb2*::*UeMsb2*, and UeT55△*UeMsb2*::*UeMsb2* were cultured on YEPS solid medium and incubated at 28 °C for 3 days. Single colonies were then selected and cultured in 10 mL of YEPS liquid medium for 2 days. The cultures were centrifuged and resuspended to an OD_600_ of 1.0. A 0.5 mL aliquot of the cell suspension was inoculated into 50 mL of YEPS liquid medium and incubated with shaking at 28 °C and 180 rpm. The OD_600_ value was measured every 12 h to generate a growth curve correlating cultivation time with OD_600_. Samples were collected every 24 h, rapidly frozen in liquid nitrogen, and stored at −80 °C. The experiment included three biological replicates and technical repeats. Additionally, cellular morphology during the culture process was observed and recorded using an XD series optical microscope (Sunnyoptical, Yuyao, China).

### 2.7. In Vitro Mating Experiment

Following the method described in Section 2.6 for growth rate measurement, each tested strain was cultured in liquid media, centrifuged, and then resuspended to an OD_600_ of 2.0. Equal volumes of sexually compatible strains were mixed, and 2 µL of the cell suspension was inoculated onto YEPS solid media. The plates were incubated in a 28 °C incubator, and every 12 h, changes in hyphal growth were observed under an SZN zoom stereo microscope (Sunnyoptical, Yuyao, China), while the formation of conjugation tubes was observed under an XD series optical microscope (Sunnyoptical, Yuyao, China). Every 24 h, colony samples were taken and frozen rapidly in liquid nitrogen for storage at −80 °C. The experiment included three biological replicates and technical repeats.

### 2.8. Stress Response Experiment

Following the method described in Section 2.6 for growth rate measurement, and referencing the relationship between the biomass of *U. esculenta* and OD_600_ values [4], each test strain was cultured in liquid media, centrifuged, and resuspended to 10^7^ cfu/mL, then serially diluted to 10^3^ cfu/mL. Different concentrations of wild-type strains, *UeMsb2* deletion strains, and *UeMsb2* complementation strains were inoculated onto YEPS solid media supplemented with 0.5 M NaCl, 0.5 M KCl, 0.1 M H_2_O_2_, 0.05 mM SDS, and 0.05 M Congo Red, and cultured in the dark at 28 °C for 3 days. Four different temperatures—15 °C, 25 °C, 28 °C, and 30 °C—were tested to verify the temperature stress response of *U. esculenta*. In addition, the wild-type strain was inoculated onto YEPS solid media containing different concentrations of NaCl and KCl (0.001, 0.003, 0.006, 0.0125, 0.025, 0.05, and 0.1 M). After 3 days of dark culture, the optimal high-osmotic concentration was observed and selected. In vitro mating experiments were conducted on YEPS solid media at the optimal high-osmotic concentration.

### 2.9. Plant Inoculation and Confocal Microscopic Observation

Following the method described in Section 2.6 for growth rate measurement, each tested strain was cultured in liquid media, centrifuged, and then resuspended to an OD_600_ of 2.0. The cell suspensions of the wild-type strains UeT14 and UeT55, the mutant strains UeT14△*UeMsb2* and UeT55△*UeMsb2*, and the complementation strains UeT14△*UeMsb2*::*UeMsb2* and UeT55△*UeMsb2*::*UeMsb2* were mixed in a 1:1 ratio. Using the method optimized by Yin (2019) [40] for artificially inoculating wild *Z. latifolia*, the mixed cell suspension was injected at the base of the wild *Z. latifolia* seedlings. At 0, 3, and 9 days post-inoculation, samples of leaf sheath and stem tip tissue near the inoculation site were collected. One part was quickly frozen in liquid nitrogen and stored at −80 °C, while another part was fixed using Carnoy’s solution. After manual slicing, the samples were stained with wheat germ agglutinin (WGA, Sigma, Shanghai, China) and propidium iodide (PI, Sigma, Shanghai, China) [41]. Observations were performed using the TCS-SP8 confocal microscope (Leica Microsystems, Wetzlar, Germany). The excitation wavelength for WGA was 495–530 nm, and for PI, it was 580–630 nm. The experiment included three biological replicates and technical repeats. Image processing was performed using Leica’s LAS-AF (Lite 4.0) software (Leica Microsystems, Wetzlar, Germany).

### 2.10. Gene Expression Analysis

The total RNA of *U. esculenta* was extracted from the previously obtained samples stored at −80 °C using a fungal RNA extraction kit (Sangon Biotech, Shanghai, China). The RNA quality was assessed through agarose gel electrophoresis. cDNA was synthesized using the HiScript^®^II 1st Strand cDNA Synthesis Kit (+gDNA wiper) (R212-01/02, Vazyme, Nanjing, China) following the manufacturer’s instructions. Primers were designed based on the known sequences of the target genes, and quantitative real-time PCR (qRT-PCR) was performed using HiScript^®^IIQ RT SuperMix for qPCR (+gDNA wiper) (R223-01, Vazyme, Nanjing, China) and following the instructions, with *β*-actin as the internal reference gene. The size of the amplicons was further verified by conventional PCR and agarose gel electrophoresis.

The qRT-PCR reaction program consisted of an initial denaturation at 95 °C for 30 s, followed by 40 cycles of denaturation at 95 °C for 30 s, annealing at 60 °C for 20 s, and extension at 72 °C for 30 s. Expression analysis for each gene was performed in triplicate. Relative gene expression was calculated using the 2^−ΔCt^ or 2^−ΔΔCt^ method, and statistical analysis was conducted using SPSS(v2.0) (SPSS Inc., Chicago, IL, USA). All primers are listed in Appendix A.

## 3. Results

### 3.1. Cloning and Expression Pattern Analysis of the UeMsb2

Sequencing analysis revealed that the cloned gene has an open reading frame of 3015 bp in length, contains no introns, and encodes a protein of 1004 amino acids. The theoretical isoelectric point of the protein is 6.11, with a molecular weight of approximately 94.54 kDa. Structural predictions identified a conserved serine-rich domain and a single transmembrane domain. Phylogenetic analysis indicated that the gene shares the highest homology with Msb2 from *U. maydis*, with a sequence identity of 69.54% (Figure 1). Consequently, the gene was named *UeMsb2* (GenBank No: MW768949). Furthermore, subcellular localization analysis revealed that the UeMsb2 protein is localized to the vacuole (Appendix A).

The expression of the *UeMsb2* gene in *U. esculenta* showed differential patterns during various stages of in vitro growth and infection. During the haploid budding growth stage, the expression level of *UeMsb2* significantly increased at 48 h (Figure 2A). However, during the mating stage of sexually compatible strains, the expression of *UeMsb2* began to rise gradually at 24 h, when conjugation tubes were formed, and significantly increased in the later stage of filamentous growth at 72 h (Figure 2B). This suggested that the gene was strongly induced during the haploid growth and filamentous growth processes, potentially related to haploid growth, hyphal formation, and filamentous development. In the early stage of host infection (3 days), the expression of *UeMsb2* also significantly rose, subsequently maintaining a high level of expression (Figure 2C), indicating that the gene played an important role in the infection process, possibly participating in the infection progression of *U. esculenta*.

### 3.2. The Deletion of UeMsb2 Did Not Affect the Cell Morphology and Growth Rate of U. esculenta

Through PCR verification of the target gene, resistance gene, and upstream and downstream fragments, as well as qRT-PCR analysis of *UeMsb2* expression levels, three *UeMsb2* deletion mutants from UeT14 and UeT55, as well as three complementation strains from UeT14 and UeT55, were successfully screened (Appendix A). Further comparison of the cell morphology and growth curves among the wild-type strain, *UeMsb2* deletion strains, and complementation strains revealed that the cell morphology and growth rate of both the *UeMsb2* deletion strains and the complementation strains were consistent with those of the wild-type strain, showing no abnormal budding phenomena (Appendix A).

### 3.3. The Deletion of UeMsb2 Affected the Density of Hyphae and Filamentous Growth in U. esculenta

To investigate the effect of *UeMsb2* deletion on in vitro mating and hyphal growth in *U. esculenta*, we used the wild-type strain as a control and compared haploid mating and hyphal growth among the *UeMsb2* deletion mutants and complementation strains under a microscope. Similar to the wild-type and complementation strains, the *UeMsb2* deletion mutants formed conjugation tubes after 24 h of culture. By 36 h, hyphae were visible, but their density was significantly reduced, indicating that *UeMsb2* deletion did not impair the formation of conjugation tubes or hyphae but inhibited mating efficiency. After 60 h of culture, the wild-type and complementation strains displayed normal filamentous growth, characterized by the formation of abundant white aerial hyphae. In contrast, the *UeMsb2* deletion mutants failed to grow and maintain white aerial hyphae (Figure 3), suggesting that *UeMsb2* deletion suppresses the in vitro filamentous growth of *U. esculenta*.

### 3.4. The Deletion of UeMsb2 Slowed the Infection Progression of U. esculenta

To further investigate whether the deletion of *UeMsb2* affected the infection process of *U. esculenta* by artificial inoculation, the results are shown in Figure 4. After 3 days of inoculation, the wild-type and complementation strains had successfully infected the leaf sheath and formed a large number of invasive hyphae, while the *UeMsb2* deletion strains formed only a small number of invasive hyphae locally (Figure 4A–C). After 9 days of inoculation, all the strains had successfully colonized the stem tip of *Z. latifolia*, but the hyphal number in the *UeMsb2* deletion strain was significantly lower than that in the wild-type and complementation strains (Figure 4B,C). After 60 days of inoculation, the plants infected by the wild-type and complementation strains began to form swollen stems, with all plants completely swollen by 90 days. The plants infected by the deletion strain only started to swell after 70 days, which was 10 days later than the wild-type and complementation strains and nearly 20% of the stems did not form galls; the peak swelling occurred between 75 days and 85 days post-infection, also ~10 days later than the wild-type and complementation strains (Figure 4D). Interestingly, there were no significant differences in the size of the swollen stems formed by the three strains, but the number of teliospores in the deletion strain was slightly reduced compared to the wild-type and complementation strains, and its color appeared darker (Figure 4E).

### 3.5. UeMsb2 Did Not Participate in Regulating the Stress Response of U. esculenta

To investigate the impact of *UeMsb2* deletion on the stress response of *U. esculenta*, we used the haploid wild-type strain as a control and tested the tolerance of *UeMsb2* deletion and complementation strains under various stress conditions. The results showed that, under osmotic, cell wall, cell membrane, temperature, and oxidative stress conditions [42], there were no significant differences in colony morphology or growth rate between the haploid *UeMsb2* deletion and complementation strains and the wild-type strain, indicating that *UeMsb2* did not participate in the stress response of *U. esculenta* during the haploid budding stage (Appendix A).

Further studies examined the tolerance of *U. esculenta* to osmotic stress during in vitro fusion, using the sexually compatible strains UeT14 and UeT55 as controls. The optimal osmotic conditions for *U. esculenta* in vitro fusion experiments were determined to be 0.0125 M NaCl and 0.0125 M KCl. Based on these conditions, we investigated the tolerance of the wild-type, *UeMsb2* deletion, and complementation strains to osmotic stress during in vitro fusion. It was found that the *UeMsb2* deletion strains exhibited the same hyphal fusion growth under osmotic stress as they did on regular medium (Appendix A), indicating that *UeMsb2* did not participate in regulating the osmotic response during the fusion process of *U. esculenta*.

### 3.6. The Deletion of UeMsb2 Suppressed the Expression of B Signaling Pathway Genes

The size of each gene fragment obtained from the quantitative real-time PCR experiments was verified through conventional PCR and agarose gel electrophoresis to ensure the correct expression of each gene (Appendix A).

To further investigate the potential mechanism by which the *UeMsb2* gene functions during hyphal growth and infection in *U. esculenta*, we analyzed the expression of *b* signaling pathway genes related to pathogenicity, including *Prf1*, *bE1*, *bW1*, *Hdp1*, *Rbf1*, *Biz1*, *Clp1*, and *Kpp6*, following *UeMsb2* deletion [1]. The results showed that during in vitro mating in *U. esculenta*, all genes, except *bE1* and *bW1*, significantly increased their expression during the filamentous growth stage (48 h) in the wild-type and complementation strains, while *bE1* and *bW1* showed significantly elevated expression at the late stage of filamentous growth (72 h). In the mutant strains, *Hdp1* exhibited significantly increased expression during the filamentous growth stage (48 h), while *bE1*, *bW1*, *Prf1*, *Biz1*, *Clp1*, and *Kpp6* showed significant upregulation at the late stage of filamentous growth (72 h). In contrast, *Rbf1* showed no notable changes (Figure 5A). During the infection process, the expression of these genes in the wild-type and complementation strains significantly increased after 3 days of infection and maintained high levels throughout the infection period. Similarly, in the mutant strain, *bE1*, *bW1*, *Hdp1*, *Rbf1*, and *Prf1* exhibited significant upregulation after 3 days of infection; however, at the early stages (3 days) of infection, the expression levels of all genes in the *UeMsb2* deletion strains were significantly lower than in the wild-type and complementation strains. *Biz1*, *Clp1*, and *Kpp6* were significantly upregulated only after 9 days of infection in the *UeMsb2* deletion strains, with expression levels markedly lower than those in the wild-type and complementation strains at the same time point (Figure 5B).

## 4. Discussion

In this study, we found that the Msb2 protein in *U. esculenta* localized to the vacuole. In *S. cerevisiae*, Nadia Vadaie tagged the extracellular and cytoplasmic domains of Msb2 with GFP and observed that Msb2-GFP primarily localized to the vacuole in the cytoplasmic region [46]. Likewise, in *U. maydis*, Msb2-mCherry accumulated in the vacuole at the appressorium pore [35], which aligns with our finding of UeMsb2 localization in the vacuole. Msb2-related proteins function as stress sensors in some fungi currently under study [28,47]. In *S. cerevisiae* and *C. albicans*, *Msb2* participates in regulating certain stress responses. For example, in *C. albicans*, *Msb2* is involved in responses to osmotic stress, cell wall stress, and temperature stress but not oxidative stress [28,30,31,48]. Similar functions are observed in *A. flavus* and *A. oligospora*, as mentioned in the introduction [32,33]. However, in *U. esculenta, UeMsb2* did not play a role in the regulation of stress responses. The *UeMsb2* mutant exhibited no changes in response to osmotic, oxidative, cell wall, or temperature stresses. Similarly, in *U. maydis*, the *Msb2* mutant showed no altered responses to osmotic, oxidative, or cell wall stresses, indicating that the function of Msb2 in *U. maydis* is uncoupled from the HOG pathway and is specific to pathogenicity [35]. Based on these findings, we propose that UeMsb2, like UmMsb2, has a non-conserved role as a stress sensor and does not regulate stress responses. Instead, its primary function lies in regulating filamentous growth and pathogenicity.

Msb2-related proteins play a crucial role in filamentous growth in many fungi [49,50]. In this study, in vitro mating experiments showed that the *UeMsb2* mutant strains had a lower hyphal density in the early stage of filamentous growth compared to the wild-type, and in the late stage, white aerial hyphae failed to grow and persist, indicating that filamentous growth ability was impaired. This is similar to findings in *C. albicans* [28]. Inhibition of filamentous growth in fungi can affect their infectivity and pathogenicity to some extent. Moreover, plant infection is a complex process, where changes in the fungal cell wall can reduce the ability to penetrate plant tissues [51]. In stress response and infection assays of *U. esculenta*, the *UeMsb2* mutant strains showed no change in cell wall integrity compared to the wild-type strains when grown on YEPS solid medium containing 0.05 mM SDS and 0.05 M Congo Red, which could explain why the mutant strains successfully infected *Z. latifolia* and proliferated during infection. Additionally, in the infection experiments, the *UeMsb2* mutant exhibited a pathogenicity defect. At 3 days and 9 days post-inoculation with the *UeMsb2* mutant, the number of invasive hyphae was significantly reduced compared to the wild-type strain. After 70 days of inoculation, the host stems began to swell, but this was delayed compared to the wild type, and the number of teliospores was slightly reduced relative to the wild type. We speculated that the impaired filamentous growth ability of the *UeMsb2* mutant strains after mating might be one reason for their pathogenicity defect. The influence of *Msb2* on infection ability and pathogenicity has been reported in various fungi. In *U. maydis*, the *Msb2* deletion mutants exhibited reduced colonization and attachment on plant surfaces [35]. In *F. oxysporum*, the *Msb2* deletion mutants displayed slow growth post-infection and a significant reduction in pathogenicity [34]. Notably, in *C. albicans*, *Msb2* did not affect pathogenicity [28]; however, Msb2 cleavage releases a huge glycosylation domain region capable of binding to antimicrobial peptides, protecting fungal cells from peptide damage [52,53]. Whether Msb2 plays a similar protective role in *U. esculenta* is also a subject worth exploring.

This study revealed that the deletion of *UeMsb2* significantly reduced the expression of *b* signaling pathway genes, including *Prf1*, *Hdp1*, *Rbf1*, *Biz1*, *Clp1*, and *Kpp6* [1], during hyphal growth and infection in *U. esculenta*. Previous research from our lab demonstrated that *UePrf1* shares high homology with *UmPrf1* and *MpPrf1* and is essential for cell mating and hyphal growth [43]. Consistent with this, the deletion of *UeMsb2* led to reduced hyphal density during in vitro mating, reflecting impaired mating capability and suggesting that *UeMsb2* may regulate *Prf1* expression to control mating and hyphal growth. Additionally, *UeMsb2* deletion caused a significant reduction in the expression of the *b* genes (*bE1* and *bW1*). Previous studies have shown that the *b* gene locus encodes the bE/bW heterodimeric transcription factor, which is critical for maintaining cell cycle arrest, filamentous hyphal growth, and subsequent infection processes [18]. *Rbf1* regulates most b-responsive genes and plays a pivotal role in filamentous growth, mating, and pathogenicity [15], while *Hdp1*, *Biz1*, and *Kpp6* act downstream of *Rbf1*, regulating G2 cell cycle arrest, appressorium formation, and host cell wall penetration, respectively [15,44]. *Clp1* is involved in fungal development, pathogenicity, and homeostasis [45]. Together, these findings suggest that *UeMsb2* may regulate the expression of *b* genes (*bE1* and *bW1*), thereby influencing downstream *b* pathway gene expression and modulating the pathogenicity of *U. esculenta*. However, the precise regulatory mechanism by which *Msb2* influences the pathogenicity of *U. esculenta* remains unclear and requires further investigation.

In this study, the deletion of *UeMsb2* resulted in attenuated pathogenicity of *U. esculenta*. The mutant strain exhibited a slight delay in inducing stem swelling in the host plant compared to the wild-type strain, and the production of teliospores was reduced. However, the mutant strain still triggered the formation of swollen stems in the host. These findings suggest that *UeMsb2* plays an important but non-essential role in the pathogenicity of *U. esculenta*. Furthermore, we speculate that UeMsb2 may cooperate with other sensors to regulate the pathogenicity of *U. esculenta*. For example, in *U. maydis*, Sho1 collaborates with the mucin-like protein Msb2 to regulate invasive growth and plant infection. In inoculation experiments, *Sho1* and *Msb2* single mutants were about twofold and threefold reduced in fungal biomass, respectively, compared with wild type, while the double mutant showed a reduction of about fivefold, and single mutants exhibited weaker pathogenicity than the double mutant [35]. In *Colletotrichum gloeosporioides*, *CgMsb2* and *CgSho1* also function complementarily to regulate pathogenicity [20]. These results underscore the notion that fungal pathogenicity is orchestrated by multiple sensors, which often collaborate to fulfill specific functions or activate downstream pathways. Therefore, it is essential to identify and characterize other sensors in *U. esculenta* to gain a comprehensive understanding of its pathogenicity mechanisms.

## 5. Conclusions

In this study, we cloned and characterized the *UeMsb2* gene (GenBank No. MW768949), revealing its induction during hyphal growth and infection processes in *U. esculenta*. Functional analysis showed that the deletion of *UeMsb2* had no significant impact on haploid growth or stress response. However, it impaired aerial hyphal growth, hyphal invasion, and proliferation of infective hyphae, which, compared to studies in *U. maydis*, was accompanied by the downregulation of genes in the *b* signaling pathway. Furthermore, the *UeMsb2* deletion mutant exhibited delayed stem swelling and a lower frequency of swollen stem formation. These findings highlight that *UeMsb2* plays a critical, though non-essential, role in the filamentous growth and pathogenicity of *U. esculenta*. This study extends our understanding of the functional diversity of Msb2, apart from its role as a fungal stress sensor. It also provides critical theoretical insights and foundational resources for elucidating the pathogenic mechanisms of *U. esculenta* and its interaction with *Z. latifolia*.

## Figures and Tables

**Figure 1 jof-10-00818-f001:**
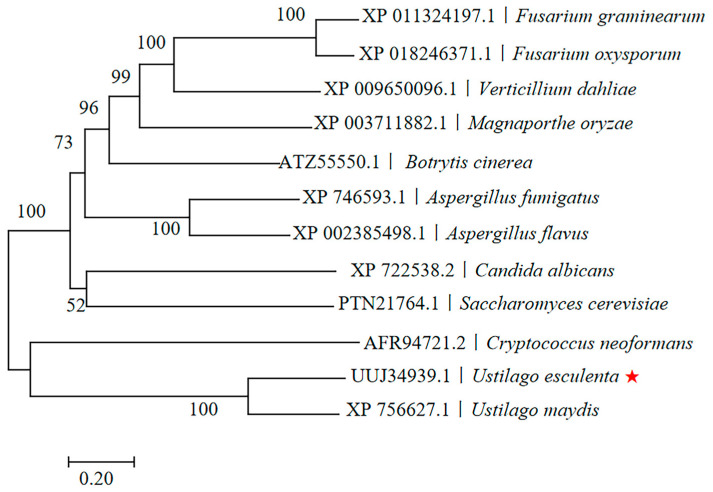
Phylogenetic analysis of *UeMsb2*. The red pentagram indicates the amino acid sequence of *UeMsb2* in the studied strain of *U. esculenta*. The phylogenetic tree was constructed using the MEGA 7.0 software, with calculations performed by the neighbor-joining method. This gene exhibited the highest homology to *Msb2* of *U. maydis*, with a sequence identity of 69.54%.

**Figure 2 jof-10-00818-f002:**
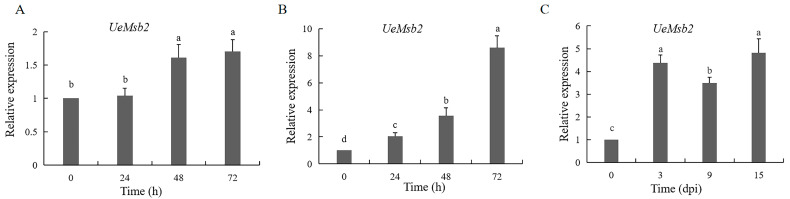
Expression pattern analysis of *UeMsb2*. (**A**) Haploid budding growth stage. (**B**) Mating and hyphal growth stages of sexually compatible strains. (**C**) Infection stage. Using the 2^−ΔΔCt^ method to calculate the relative gene expression level, the gene expression level at “0 h” was taken as “1”. Single factor analysis of variance was used to analyze the data. Different letters indicate significant differences (*p* < 0.05). The internal reference gene was *β-actin*, *n* = 3.

**Figure 3 jof-10-00818-f003:**
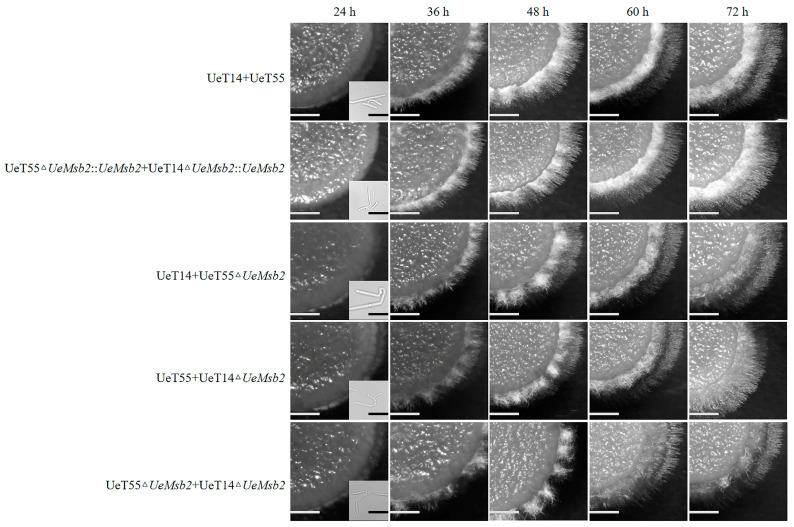
The deletion of *UeMsb2* affected the growth of aerial mycelium of *U. esculenta*. Colony morphology chart: bar=1500 μm. Conjugation tubes in the lower right corner of the images of 24 h: bar = 20 μm. UeT14+UeT55 represents the mating of wild-type strains, and UeT14△*UeMsb2*::*UeMsb2*+UeT55△*UeMsb2*::*UeMsb2* represents the mating of gene complementation strains of UeT14 and UeT55. UeT14+UeT55△*UeMsb2* represents the mating of the UeT55 gene deletion strain with the UeT14 strain, UeT14△*UeMsb2*+UeT55 represents the mating of the UeT14 gene deletion strain with the UeT55 strain, and UeT14△*UeMsb2*+UeT55△*UeMsb2* represents the mating of gene deletion strains of both UeT14 and UeT55.

**Figure 4 jof-10-00818-f004:**
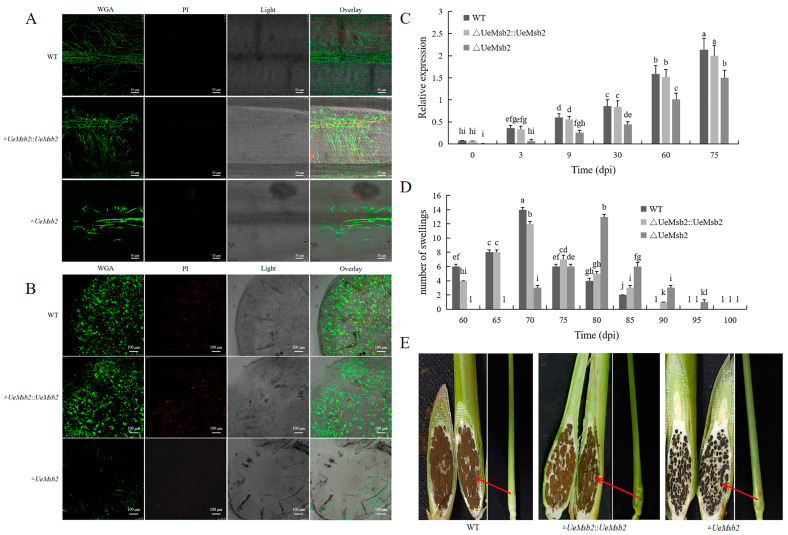
The deletion of *UeMsb2* slowed down the infection process of *U. esculenta*. (**A**) Confocal microscopy of *U. esculenta* in the leaf sheath at 3 days post-inoculation. Scale bar = 50 μm. (**B**) Confocal microscopy of *U. esculenta* in the stem tip at 9 days post-inoculation. Scale bar = 100 μm. (**C**) Relative quantification of *U. esculenta* in *Z. latifolia*. *Zl-actin* was used as the internal reference gene. The data were analyzed using two-way ANOVA. Different lowercase letters indicate significant differences (*p* < 0.05), *n* = 3. (**D**) Timeline for stem gall formation. The data were analyzed using two-way ANOVA. Different lowercase letters indicate significant differences (*p* < 0.05), *n* = 3. (**E**) Enlarged stem and longitudinal section of the plant at 70 days post-inoculation (all three were swollen plants on day 70). The red arrow in (**E**) indicates the teliospores of *U. esculenta*. WT represents the wild-type strains UeT14+UeT55, △*UeMsb2* represents the gene deletion strains UeT14△*UeMsb2*+UeT55△*UeMsb2*, △*UeMsb2*::*UeMsb2* represents the gene complementation strains UeT14△*UeMsb2*::*UeMsb2*+UeT55△*UeMsb2*::*UeMsb2*.

**Figure 5 jof-10-00818-f005:**
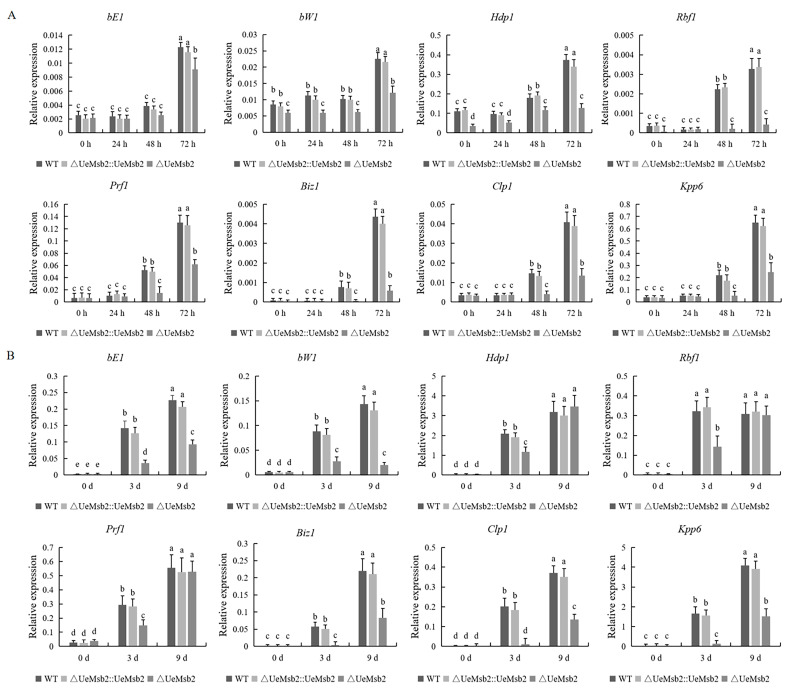
The deletion of *UeMsb2* in *U. esculenta* resulted in significantly reduced expression of *b* signaling pathway genes during in vitro mating (**A**) and infection processes (**B**). *Prf1* is essential for cell mating and hyphal growth [43]; *bE1* and *bW1* are involved in maintaining cell cycle arrest, filamentous hyphal growth, and subsequent infection processes [18]; *Rbf1* plays a role in regulating filamentous growth, mating, and pathogenicity [15]; *Hdp1*, *Biz1*, and *Kpp6* are responsible for regulating G2 cell cycle arrest, appressorium formation, and penetration of the host plant cell wall, respectively [15,44]; *Clp1* is involved in regulating fungal development, pathogenicity, and fungal homeostasis [45]. *β-actin* was used as the internal reference gene. The data were analyzed using two-way ANOVA. Different lowercase letters indicate significant differences (*p* < 0.05), *n* = 3.

## Data Availability

The data supporting this study’s findings are available from the author, Yafen Zhang, upon reasonable request.

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
