# Peer review of "Role of UeMsb2 in Filamentous Growth and Pathogenicity of Ustilago esculenta"

_jof, 2024, doi:10.3390/jof10120818_

Round 1
Reviewer 1 Report
In general, the manuscript of Wanlong Jiang and coauthors is clearly and logically written, the design of the experiments and the results obtained are adequately presented. The work can be published after some revision. English language correction is also required.
1. In my opinion, you can shorten the introduction by removing information that is not directly related to the work.
2. Check the use of abbreviations throughout the text, starting with the introduction. Explanations must be given everywhere at the first mention.
3. Question about Msb2 protein/gene homology. Still, how similar they are in different plants? And also, whether they have different names. So, the authors use UeMsb2 for Ustilago esculenta, but similar proteins/genes from different plants are called Msb2.
4. Regarding Gene Expression Analysis. Where did the primer structures come from? What was the size of the resulting fragments in real time PCR experiments? How did authors check the quality of RNA and amplicons size? Provide data in the supporting materials.
5. Add statistics and significance to Figure 4D.
6. Сheck/correct line 71, the sentence lines 412-413, rephrase line 491.
7. Slang expressions should be removed (line 456).
8. Regarding the assumption that UeMsb2 is involved in the regulation of b signaling pathway genes. What is known in this context about other similar proteins/genes?
9. Regarding the conclusions. They should be corrected. Lines 495-497 - possibly due to regulation of gene expression? Lines 498-500 - the authors did not reveal the participation of UsMsb2 in stress.
Reviewer 2 Report
This is mainly a confirmatory study of Msb2 mucin in a plant pathogenic fungus. Experiments are appropriately designed and figures are reasonably organized.
My major concerns are: 1) This manuscript is not well written. It needs a major revision. 2) Conclusions need to be toned down. Based on data presented in the manuscript, Msb2 is important but not essential.
My major concerns are: 1) This manuscript is not well written. It needs a major revision. 2) Conclusions need to be toned down. Based on data presented in the manuscript, Msb2 is important but not essential.
Reviewer 3 Report
The manuscript explores the role of the UeMsb2 gene in Ustilago esculenta, which infects Zizania latifolia, causing stem swelling and leading to the formation of edible fleshy stems known as jiaobai. The study presents evidence that UeMsb2, as part of the MAPK signaling pathway, is critical in regulating filamentous growth and pathogenicity but not stress responses in U. esculenta. Through gene deletion and complementation studies, the authors demonstrate that UeMsb2 influences mating, hyphal growth, and host infection progression. These findings contribute to understanding the pathogenic mechanisms of U. esculenta.
Overall, the manuscript is a valuable contribution to fungal pathogenesis research. The findings support UeMsb2's role in U. esculenta pathogenicity and provide a basis for further studies on MAPK pathway signaling in plant-fungal interactions. Minor revisions as suggested would improve the clarity and impact of the manuscript.
The deletion experiments convincingly show UeMsb2's role in mating and hyphal growth, supported by clear figures. Quantitative data on the impact of UeMsb2 on the density and formation of hyphae during infection would strengthen claims.
The study highlights the unaffected stress response in UeMsb2 mutants, implying specific functionality in filamentous growth regulation. This finding is interesting and should be emphasized more in the discussion.
Round 2
Reviewer 1 Report
The article may be accepted for publication.
The authors answered all my comments/questions and made the necessary corrections/additions to the text of the manuscript and supplementary materials.
Author Response
Comments 1: The article may be accepted for publication. The authors answered all my comments/questions and made the necessary corrections/additions to the text of the manuscript and supplementary materials.
Response 1: Thank you very much for your thoughtful review and for acknowledging the revisions we made in response to your comments and suggestions. We truly appreciate your encouraging feedback and are pleased that the changes have addressed your concerns. We are hopeful that the manuscript meets the standards for publication and are grateful for the opportunity to further improve it through your guidance. Thank you again for your time and valuable insights.